# IN REINFORCEMENT LEARNING, ALL OBJECTIVE FUNCTIONS ARE NOT EQUAL

**Romain Laroche & Harm van Seijen**
Microsoft Research Montréal, Canada
romain.laroche@microsoft.com

## ABSTRACT

We study the learnability of value functions. We get the reward back propagation out of the way by fitting directly a deep neural network on the analytically computed optimal value function, given a chosen objective function. We show that some objective functions are easier to train than others by several magnitude orders. We observe in particular the influence of the $\gamma$ parameter and the decomposition of the task into subtasks.

## 1 INTRODUCTION

Most of Reinforcement Learning (RL, Sutton & Barto, 1998) research is focused on the training process: how to propagate the reward information across the states, in order to implement planning capabilities into the agent. Deep Reinforcement Learning (Mnih et al., 2013) relies on experience replay and other tricks to allow steady training of the optimal value function: $V^*$. Only very recently, some papers started to tackle the question of $V^*$ learning efficiency (Xu et al., 2017; Lehnert et al., 2018). This extended abstract proposes an experimental study along these lines.

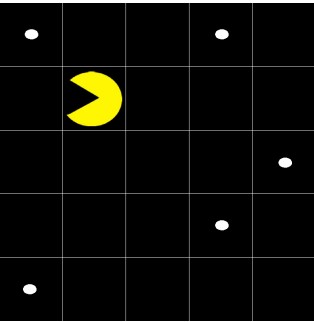

Figure 1: Fruit collection domain

We intend to show that the value function is easier to learn depending on the choice of the objective function. We consider a fruit collection task where the agent has to navigate through a $5 \times 5$ (5 fruits are randomly placed at the beginning of each episode, see Figure 1). This is small enough to be a solvable Travelling Salesman Problem and large enough to count more than 1,000,000 possible states:

$$nb_{states} = nb_{pos} \sum_{nb_{fruits}=0}^{nb_{max}} \binom{nb_{pos}-1}{nb_{fruits}} = 25 \sum_{k=0}^{5} \binom{24}{k} = 1,386,375. \tag{1}$$

A deep neural network (DNN) is fitted from a limited amount of samples to the ground-truth value function $V_\gamma^{\pi^*}$ for various objective functions: *TSP-obj* is the optimal number of turns to gather all the fruits, *RL-obj* is the optimal $\gamma$-discounted return, and *DEC-obj* is the decomposed return as defined in Laroche et al. (2017); Van Seijen et al. (2017). We also study the effect of the $\gamma$ discount factor hyperparameter on the learnability of the function. All the trained DNNs are then evaluated on the Travelling Salesman Problem criteria: the time to gather all the fruits. To evaluate their performance, actions are selected greedily by moving the agent up, down, left, or right to the neighbouring grid cell of highest value.

Our results show that the more complex the objective function is, the most difficult it is to train the DNN. In particular, we show that *TSP-obj* is extremely difficult to train and that the *DEC-obj* approach is faster than *RL-obj* to train by a magnitude order. It also shows that low $\gamma$ values yield poor results as expected, because the value function tends to be very close to zero, but more surprisingly, classically chosen high $\gamma > 0.9$ values are also suboptimal. On this task, $\gamma$ values around 0.7 or 0.8 seem to be fastest to train bot h on *RL-obj* and *DEC-obj*.

## 2    EXPERIMENTAL SETTING

All trainings are performed from the same state and the same network: similarly to the Taxi Domain (Dietterich, 1998), we incorporate the location of the fruits into the state representation by using a 50 dimensional bit vector, where the first 25 entries are used for fruit positions, and the last 25 entries are used for the agent's position. The DNN feeds this bit-vector as the input layer into two dense hidden layers with 100 and then 50 units. The output is a single linear head representing the state-value, or a multiple head in the case of the vector *DEC-obj* target. In order to assess the value function complexity, we train for each discount factor setting a DNN of fixed size on 1000 random states with their ground truth values. Each DNN is trained over 500 epochs using the Adam optimizer (Kingma & Ba, 2014) with default parameters.

We tried to train the network on each objective function with an unlimited number of samples and we found that *TSP-obj* and *RL-obj* were able to largely surpass the optimal *DEC-obj* performance. The difference we are going to observe are therefore not explained by a lack of representation capacity in the DNN. In our further experiments, this learning problem is fully supervised on 1000 samples, allowing us to show how sample-wise efficient a DNN can capture $V^*$ while ignoring the burden of finding the optimal policy and estimating its value functions through TD-backups or value iteration.

The training difference solely lies in the four objective function targets that are considered:

- The *TSP-obj* target is the natural objective function, as defined by the Travelling Salesman Problem: the number of turns to gather all the fruits:

$$y_{TSP}(x) = - \min_{\sigma \in \Sigma_k} \left[ \sum_{i=1}^{k} d(x_{\sigma(i-1)}, x_{\sigma(i)}) \right],$$

  where $k$ is the number of fruits remaining in state $x$, where $\Sigma_k$ is the ensemble of all permutations of integers between 1 and $k$, where $\sigma$ is one of those permutations, where $x_0$ is the position of the agent in $x$, where $x_i$ for $1 \le i \le k$ is the position of fruit with index $i$, where $d(x_i, x_j)$ is the distance ($||\cdot||_1$ in our gridworld) between positions $x_i$ and $x_j$.

- The *RL-obj* target is the objective function defined for an RL setting, which depends on the discount factor $\gamma$:

$$y_{RL}(x) = \max_{\sigma \in \Sigma_k} \left[ \sum_{i=1}^{k} \gamma^{\sum_{j=1}^{i} d(x_{\sigma(j-1)}, x_{\sigma(j)})} \right],$$

  with the same notations as for TSP.

- The summed *DEC-obj* target does not involve the search into the set of permutations and can be considered simpler to this extent:

$$y_{ego}(x) = \left[ \sum_{i=1}^{k} \gamma^{d(x_0, x_i)} \right],$$

  with the same notations as for TSP.

- The vector *DEC-obj* target is the same as the summed one, except that the target is now a vector, separated into as many channels as potential fruit position:

$$\boldsymbol{y}_{ego}(x) = \begin{cases} \gamma^{d(x_0, x_i)} & \text{if there is a fruit in } x_i, \\ 0 & \text{otherwise.} \end{cases}$$

## 3    RESULTS AND DISCUSSION

Figure 2 displays the performance of the theoretical optimal policy for each objective function in dashed lines. The theoretical optimal policy can be searched with brute force (the problem is small enough to allow it). We compute the optimal value of the state by taking the value of the best order of fruit collection. *TSP-obj* and *RL-obj* targets largely surpass the *DEC-obj* one. However, the networks trained on the limited data of 1000 samples yield completely different result (shown in solid lines).

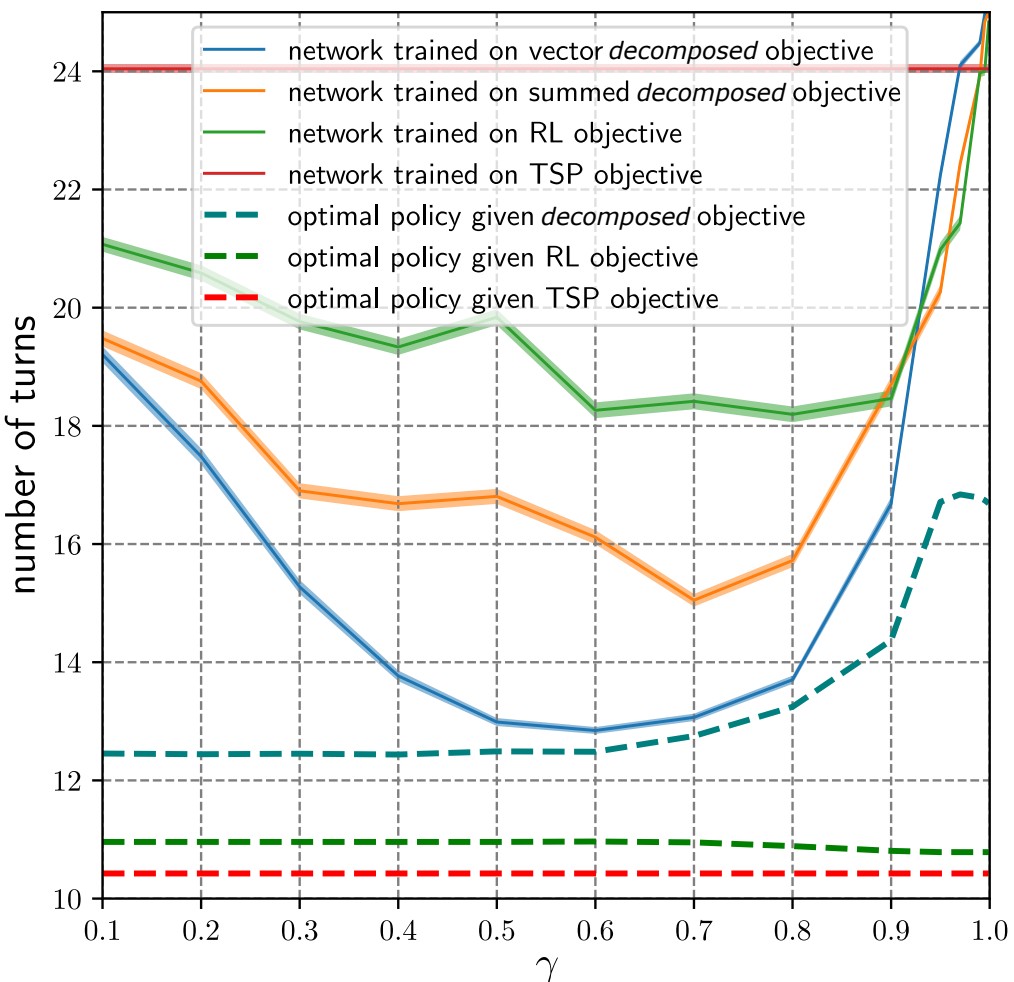

Figure 2: Value function training.

The *TSP-obj* target is the hardest to train on. The *RL-obj* target follows as the second hardest to train on. The *Dec-obj* target is easier to train on, even without any state space reduction, or even without any reward/return decomposition (summed version). Additionally, if the target value is decomposed (vector version), the training is further accelerated, which was to be expected since the signal is richer. Finally, we found that the *DEC-obj* performance tends to dramatically decrease when $\gamma$ gets close to 1, because of attractors' presence (Laroche et al., 2017). We consider this small experiment to show that the complexity of objective function is critical and that decomposing it may make it simpler and therefore easier to train by a factor 10, even without any state space reduction.

We also observe the value function learnability dependency on $\gamma$ that was reported in previous works (Petrik & Scherrer, 2009; Lehnert et al., 2018). The performance curves of the policies greedy with respect to the trained DNNs present a pronounced U-shape. For small $\gamma$ values, the range of the value function (and thus the action gap) collapses quickly as one moves away from fruit locations. In this case, both models cannot reach a high enough precision and hence perform worse for low $\gamma$ values. Further, for very high $\gamma$ values action-gaps collapse, because the Fruit Collection Task contains terminal states, which we believe also results in reduced performance of both DNNs.

If those preliminary results do not allow us to give general recommendations, they underline the importance of the choice of the objective function, and the fact that, it is sometimes preferable to aim at sub-optimal solutions that are easier to train.

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
