# OpenReview forum: "In reinforcement learning, all objective functions are not equal"
_ICLR.cc/2018/Workshop — Accept_

### Official Review · AnonReviewer1 · 2018-03-10
**Interesting experiment on the choice of objective functions for RL**

**Rating:** 7
**Confidence:** 3

**Review:**

In this paper the authors study the effect of the choice of the objective function in the ease of training Reinforcement Learning agents to solve the particular task. In particular the authors base their experiments on a toy task of fruit gathering on a 5x5 grid which is very similar to the Taxi environment. The goal is to find the shortest path navigating the grid to collect all the fruit, and is small enough to be a solvable TSP instance. The authors then train RL agents on a variety of objective functions including the TSP objective, the RL objective which is the cumulative discounted term, and the decomposed reward. The authors also study the effect of the choice of the discount factor on the ease of learnability of the problem. The authors conclude that the more complicated the objective function (like the TSP objective function), the harder it is to train. The authors also make the observation that very high values of the discount factor (> 0.9) are also harder to train.

---

### Official Review · AnonReviewer4 · 2018-03-18
**Interesting Empirical results, but lacks related work analysis / references**

**Rating:** 5
**Confidence:** 3

**Review:**

Empirical paper which shows somewhat surprising results: suboptimal reward functions may lead to better solutions due to learnability of the underlying (potentially suboptimal) reward shaping.

The paper uses TSP in an RL context, and it should be updated to contain several related works which demonstrate the effectiveness of RL / supervised learning in the context of TSP. Ideally, it should use the same setup as in [1,2], which would make the contributions much stronger and interesting.

[1] https://arxiv.org/abs/1611.09940
[2] https://arxiv.org/abs/1506.03134

---

### Decision · Program_Chairs · 2018-03-20
**ICLR 2018 Workshop Acceptance Decision**

**Decision:**

Accept

**Comment:**

Congratulations, your paper was accepted to the ICLR workshop.